# Dietary and nutritional interventions in children with cerebral palsy: A systematic literature review

**Fernanda Rebelo**[1]*, **Isabela Rodrigues Mansur**[1,2], **Teresa Cristina Miglioli**[1¤], **Maria Dalva Baker Meio**[1], **Saint Clair Gomes Junior**[1]

1 Clinical Research Unit, National Institute of Women, Children and Adolescents Health Fernandes Figueira (IFF), Oswaldo Cruz Foundation (Fiocuz), Rio de Janeiro, RJ, Brazil, 2 Undergraduate Program in Nutrition, Fluminense Federal University (UFF), Niterói, RJ, Brazil

¤ Current address: Nutrition Department, Iguaçu University (UNIG), Nova Iguaçu, RJ, Brazil
* frebelos@gmail.com

**Data Availability Statement:** All relevant data are within the paper and its Supporting Information files.

## Abstract

### Background

Cerebral palsy is an extremely severe brain injury associated with multiple nutritional and clinical issues, such as underweight, gastroesophageal reflux, constipation, and nutrient deficiency. Evidence-based dietary and nutritional interventions may improve the quality of life of children with cerebral palsy.

### Aim

Systematically review randomized clinical trials evaluating nutritional and dietary interventions in the clinical, nutritional, and neurodevelopmental aspects of children with cerebral palsy.

### Methods

A search was performed in electronic databases (LILACS, Medline, Web of Science, Embase, Scopus, Cochrane Library, ClinicalTrials.gov, Brazilian Digital Library of Theses and Dissertations, ProQuest Dissertations and Theses Database, OpenGrey) using keywords. The search was firstly performed in May 2020 and updated on June 18th, 2021. Eligible studies were randomized clinical trials, that included children between 2 and 12 years old, and evaluated the effect of nutritional or dietetic interventions on clinical, nutritional or neurodevelopmental outcomes. Risk of bias was investigated using the RoB-2 tool. The study was registered on PROSPERO (CRD42020181284).

### Results

Fifteen studies were selected. Positive results included the use of whey-based or pectin-enriched enteral formulas for gastroesophageal reflux (n = 6); 25-hydroxy-vitamin D supplementation for hypovitaminosis D (n = 2); supplementation with lipid mixture or diet with high-density energy for improvements in anthropometric measures (n = 2); supplementation with

**Funding:** This study was funded by the National Research Council (CNPq, process number 442551/2019-3). The funders had no role in the study design, data collection and analysis, decision to publish, or preparation of the manuscript.

**Competing interests:** The authors have declared that no competing interests exist.

probiotics, prebiotics, symbiotics or magnesium for constipation (n = 2); nutritional support system for gross motor function (n = 1); lactoferrin and iron hydroxide polymaltose for iron deficiency anemia (n = 1); and educational intervention to improve feeding skills (n = 1). The overall risk of bias was high for 60% of the studies, and some concerns were raised for the remaining 40%.

## Conclusion

Some promising dietary and nutritional interventions may promote important clinical improvements for patients with cerebral palsy. However, evidence is weak, as few clinical trials have been published with many methodological errors, leading to a high risk of bias.

## Introduction

Cerebral palsy (also known as chronic non-progressive encephalopathy) is an extremely severe, disabling brain injury with repercussions that are not limited to the individual, affecting their families and society. This health condition comprises a group of permanent and non-progressive disorders of movement and posture secondary to injury, dysfunction, or damage to fetal or infant brain development. These motor disorders are often accompanied by changes in sensory, perception, cognition, communication, behavior, epileptic seizures, secondary musculoskeletal problems, and important comorbidities [1,2]. These comorbidities include nutritional issues, such as underweight, gastroesophageal reflux, constipation, and deficiency of micro- and macronutrients [3–6].

The influence of nutrition on changes in the functioning of the human body is recognized, both in the causal chain of diseases and in the modification of responses in already established diseases. Specific nutrients can act in enzymatic processes, such as zinc and magnesium, or in the formation of hemoglobin, such as iron, in bone formation, such as vitamin D and calcium, in the formation of muscle mass, such as proteins [7–9]. The use of alternative enteral formulas or addiction of fibers, probiotics, prebiotics, lipids or fatty acids in the diet can act as an adjunct in the treatment of gastroesophageal reflux, constipation, undernutrition and others [10–12]. Therefore, nutritional interventions can act in several ways to improve the quality of life of patients with cerebral palsy.

The identification of dietary or nutritional interventions with good results for the treatment of common disorders in children with cerebral palsy, with high-level scientific evidence, is important for the construction of protocols and for the guidance of evidence-based practice. It provides subsidies for the management and organization of nutritional care in public health, promoting the reduction of morbidity and hospital costs and the improvement of the quality of life of children with cerebral palsy. Thus, the present systematic review aims to synthetize the results of randomized clinical trials evaluating the effect of nutritional and dietary interventions in clinical, nutritional and neurodevelopmental outcomes of children with cerebral palsy.

## Methods

This is a systematic literature review (SLR) conducted following the recommendations of the Cochrane handbook for systematic review of interventions and reported according to the Preferred Reporting Items for Systematic reviews and Meta-Analyses (PRISMA) Statement

[13,14] (PRISMA checklist is available in S1 Appendix). This SLR was registered on PROS-PERO under the number CRD42020181284 (register details are available in S2 Appendix). Our aim is to determine the effect of nutritional and dietary interventions on clinical, nutritional and neurodevelopmental outcomes in children with cerebral palsy.

## Search strategy

We performed an automatic literature search in the following electronic databases: LILACS, Medline, Web of Science, Embase, Scopus, Cochrane Library, ClinicalTrials.gov, Brazilian Digital Library of Theses and Dissertations, ProQuest Dissertations and Theses Database, and OpenGrey. The following keywords were used and adapted for each database: ("cerebral palsy" OR "chronic non progressive encephalopathy") AND (child OR children OR childhood) AND (nutrients OR nutrition OR diet). Moreover, sensitive strategies were applied to identify randomized controlled trials. A search summary with a complete strategy for each database can be found in S3 Appendix. Finally, we performed a backward search, manually checking the reference lists of the eligible studies. Our literature search was first performed in May 2020 and updated on June 18th, 2021.

To be included in the SLR, the studies had to be randomized clinical trials, include children between 2 and 12 years old, and evaluate the effect of nutritional or dietetic interventions on clinical, nutritional, neurodevelopmental and/or seizure control outcomes. There were no restrictions regarding year, idiom or type of publication.

## Studies selection

The retrieved references were managed using Covidence software [15]. First, titles and abstracts were screened. Second, the full texts were checked for eligibility. Both steps were performed by two independent reviewers (IRM, TCM). Disagreements were resolved by a third investigator (FR). When multiple reports of the same study/database were identified, the more detailed report was selected for the SLR. When the reference was a register (in ClinicalTrials.gov or Cochrane Library) with no publications associated, we searched for the principal investigator productions to obtain the results.

## Data extraction

Data extraction was performed using a structured questionnaire by two authors (IRM, TCM). A third author (FR) reviewed the data and resolved any disagreements. The following information was extracted from the selected studies: year of publication, country and setting where the study was conducted, follow-up period, sample size, recruitment and randomization process, losses of follow-up, inclusion and exclusion criteria, sample characteristics (age, gender, functional mobility), intervention tested, control group, blindness, time from intervention to outcome evaluation, methods for outcome evaluation, statistical analysis and summary of the main results.

## Data synthesis

It was not possible to perform a quantitative synthesis due the clinical and methodological heterogeneity of the included studies. For the meta-analysis, at least two studies investigating the same intervention and outcome would be necessary, which did not occur. We performed a structured narrative synthesis based on the Guidance on the Conduct of Narrative Synthesis in Systematic Reviews [16]. The approaches chose for the synthesis, considering the nature of our data, were: textual descriptions of studies; groupings and clusters; and tabulation.

## Risk-of-bias assessment

The risk-of-bias assessment was performed using the revised Cochrane tool for assessing risk of bias in randomized trials (RoB 2 tool) [17] in a study level. RoB 2 is a structured tool that evaluate five domains of bias. Each domain, formed by a series of questions, can be judgement as 'Low' or 'High' risk of bias, or can express 'Some concerns'. The assessment was performed by two independent investigators (IRM, TCM). Differences were solved by a third reviewer (FR).

To synthetize the data regarding risk of bias, a summary risk of bias table was generated using the Risk-of-bias VISualization (robvis) tool [18]. Results are described as percentual of studies with 'High risk' or 'Some concerns' for each domain. The domains judged as high risk of bias were discussed to provide reasons for this judgement and recommendations for futures studies.

The complete RoB 2 evaluation is presented as supplemental material (S1 Table). Studies were classified with an overall high risk of bias when at least one domain was judged as having a high risk of bias or when some concerns were found in at least three domains.

## Results

A total of 508 references were obtained from electronic databases. After exclusion of duplicates, 399 references were screened against titles and abstracts, and 43 full-text studies were assessed for eligibility. Fifteen studies met all eligibility criteria and were selected for the SLR. A flowchart with a detailed selection process is described in Fig 1. The references excluded after full-text evaluation and the respective reasons are listed in S2 Table.

We observed a preponderance of studies performed in North America (3 in Canada, 2 in Mexico, 1 in the United States), followed by Africa (2 in Egypt, 1 in Tanzania), South America (1 in Bolivia, 1 in Chile), Europe (1 in Finland), Asia (1 in Japan) and Oceania (1 in Australia). The first record is the study from Patrick and colleagues from Canada in 1986 [19]. In the following twenty years publications were scarce, with only 4 studies. In 2007 publications became more frequent, with 11 studies in the last 14 years (Tables 1–5).

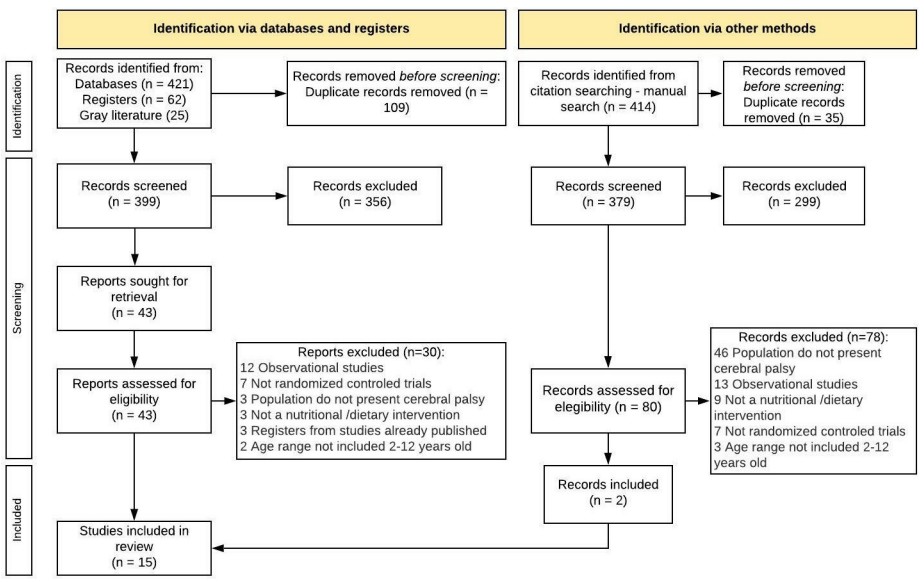

**Fig 1. Flow-chart illustrating the search and selection process.**

Considering the main outcomes investigated, studies were classified into 5 groups: (1) gastric emptying rate, gastroesophageal reflux (GER) and related symptoms; (2) plasma concentration of 25-hydroxyvitamin D; (3) anthropometric measures and nutritional status; (4) constipation and stool characteristics; and (5) other outcomes evaluated by only one study (Tables 1–5). A summary of statistic methods used and main quantitative results for each study can be found in S4 Appendix.

The group of studies with main outcomes involving gastric emptying, GER and related symptoms comprised six works [12,20–24]. Five of the six studies [12,20,22–24] were crossover trials that modified the source of protein in the enteral formulas, comparing different proportions of whey and casein. In summary, formulas with higher proportions of whey seem to accelerate gastric emptying, which may contribute to the reduction of GER and diminish symptoms like gagging and retching. However, one study [22] observed higher pain scores in children receiving formula 100% whey than in those receiving formula 50% whey and 50% casein, while another study [12] reported a relation between rapid gastric emptying and postprandial symptoms such as nausea, diarrhea, sweating and retching. One study [21] evaluated the use of enteral formulas with high (2:1 v/v) or low pectin (3:1 v/v) for 4 weeks, finding a significant reduction in GER compared to a formula with no pectin (Table 1).

Two studies investigated the effect of supplementation with vitamin D on the plasma concentration of 25-hydroxyvitamin D, with a total of 63 participants. Age varied between 6 and 18 years old. The study of LeRoy et al. (2015) evaluated a single oral dose of 100,000 UI compared to a placebo [7]. Kilpinen-Loisa et al. (2007) investigated the dose of 1,000 UI, 5 days/week orally for 10 weeks compared to an observational group (no placebo) [25]. Both studies found improvements in 25-hydroxy-vitamin D plasma concentrations for children receiving intervention compared to control groups. Moreover, the interventions were not associated with hypercalcemia or any other adverse effects (Table 2). Besides the plasma concentration of 25-hydroxy-vitamin D, Kilpinen-Loisa et al (2007). also investigated levels of calcium, phosphate, parathyroid hormone and markers of bone formation and resorption (alkaline phosphatase, serum aminoterminal propeptide of type I procollagen and telopeptide of type I collagen). The authors did not find significant differences in these markers of bone metabolism between groups who received vitamin D supplementation or not.

Two studies evaluated anthropometric measures and nutritional status as the main outcomes, comprising 24 participants. One study included only children below 5 years of age, and the other had a sample composed of individuals between 2.8 and 15.8 years old. Sevilla Paz Soldán et al. (2018) investigated supplementation with a lipid mixture (coconut 35%, olive 35%, fish 15%, soy 15%) for 6 months [11]. Patrick et al. (1986) evaluated a diet with high energy density for 5 weeks [19]. Both interventions resulted in significant improvement of anthropometric measures, such as weight, arm circumference and triceps skinfold. Moreover, the lipid mixture was associated with a better lipid profile and psychomotor development at the end of the intervention compared to the control group (Table 3).

Two studies applied interventions to improve constipation and stool characteristics. Age ranged between 1 and 12 years old for the 136 participants of both studies. García-Contreras et al. (2020) [10] evaluated supplementation for 28 days with probiotics, symbiotics or prebiotics compared to a placebo. Hassanein et al. (2021) [26] studied supplementation with an oral solution of magnesium sulfate for 1 month compared to a placebo. Both studies found positive results of their interventions, with improvements in stool consistency and frequency (Table 4).

Three studies evaluated other outcomes, comprising gross motor function, iron deficiency anemia and feeding skills. Aiming to improve gross motor function, Leal-Martinez et al (2020) [27] compared a nutritional support system with the diet recommended by the World Health Organization (WHO) and an observational group of patients between 4 and 12 years old. The

**Table 1. Characteristics of the studies evaluating as main outcome gastric emptying rate, gastroesophageal reflux and related symptoms.**

| Study | Country | N | Age (Years) | Sex F (%) | Sex M (%) | Functional mobility | Groups (n) | Interventions | Duration of treatment | Outcome studied | Summary of key results |
|---|---|---|---|---|---|---|---|---|---|---|---|
| Fried, 1992 [23] | Canada | 9* | 3–18 | NA | NA | Spastic quadriplegia | ITV (9) | Enteral formulas (A) Whey predominant [60% whey + 40% casein], (B) 100% Whey hydrolysate, (C) 100% Whey hydrolysate with 70% of the fat as MCT | 2 hours for gastric emptying and 1 month for vomiting | Gastric emptying, episodes of vomiting | Whey-based formulas led to a faster gastric emptying and reduced the number of episodes of vomiting. The faster gastric empty rate was obtained with formula B. |
| | | | | | | | Control (9) | Enteral formula casein predominant (80% casein, 20% soy) | | | |
| Khoshoo, 1996 [20] | USA | 10* | 4.5–14.5 | 40 | 60 | NA | ITV (10) | Whey-based enteral isosmolar formula (4g protein, 100% whey) | 48 hours | GER | The whey-based diet significantly reduced the frequency and duration of GER episodes when compared to the casein-based diet. |
| | | | | | | | Control (10) | Casein-based isocaloric and isosmolar enteral formula (3.7g protein, 84% casein, 16% soy) | | | |
| Graham-Parker, 2001 [24] | Canada | 20* | 2–9 | NA | NA | NA | ITV (20) | Enteral formulas (A) 18% whey; (B) 50% whey | 4 weeks | Emesis, gagging/ retching, stool frequency and consistency, volume intakes, degree of irritability and anthropometric parameters. | Diet B caused less gagging/retching than Diet A. There was a slight decrease in stool frequency during the Diet B period as compared to the Diet A period. Stool consistency remained the same with Diet B and became softer than usual during Diet A period |
| | | | | | | | Control (20) | The formulas were compared with each other | | | |
| Savage, 2012 [22] | Australia | 13* | 2.4–15.4 | 38.5 | 61.5 | Spastic quadriplegia: 53.8% Spastic Diplegia: 23.1% Hypotonia: 15.4% Dyskinetics: 7.7% | ITV (13) | Enteral formula 50% whey/ 50% casein; or 100% enteral formula partially hydrolyzed whey | 1 week | GER, gastric emptying and symptoms of food intolerance (choking, regurgitation, irritability, regurgitation and pain) | Whey-based formulas led to faster gastric emptying, compared to casein-based formula. Reflux parameters remained unchanged. GI symptoms were less frequent in children who received the formula 50% whey, compared to those who received 100% whey, whose pain scores worsened. |
| | | | | | | | Control (13) | Standard casein-based enteral formula (82% casein and 18% serum) | | | |

*(Continued)*

**Table 1.** (Continued)

| Study | Country | N | Age (Years) | Sex F (%) | Sex M (%) | Functional mobility | Groups (n) | Interventions | Duration of treatment | Outcome studied | Summary of key results |
|-------|---------|---|-------------|-----------|-----------|---------------------|------------|---------------|----------------------|-----------------|------------------------|
| Brun, 2012 [12] | Norway | 15* | 4–15 | 46.7 | 53.3 | NA | ITV (15) | Enteral formulas (A) casein 100%; (B) Hydrolyzed whey; (C) Amino acids; or (D) 40% casein/ 60% whey | 4 days | Gastric emptying rate; Postprandial GI symptoms | The shortest gastric emptying time was obtained when the children received the formula D: 40% casein/ 60% serum. For formula D, emptying was significantly faster in children with postprandial symptoms, compared to those without symptoms. |
| | | | | | | | Control (15) | The formulas were compared with each other | | | |
| Miyazawa, 2008 [21] | Japan | 18 | 11.7 ± 4.4 | 11.1 | 88.9 | NA | ITV 1 (9) | Enteral formula rich in pectin [liquid pectin = 2: 1 (v / v)] | 4 weeks | GER (esophageal pH) and symptoms of GER disease (vomiting, residual gastric volume, wheezing and cough episodes, use of oxygen for dyspnea) | The diet rich in pectin significantly reduced the GER index, the number of GER episodes per day, the duration of the GER and the number of vomit episodes, compared with the diet without pectin. Both the diet rich in pectin and low pectin reduced the cough score, compared to the diet without pectin. |
| | | | | | | | Control (9) | Enteral formula with low [liquid pectin = 3: 1 (v / v)] or no pectin | | | |

Note: * Cross-over study. GER, Gastroesophageal reflux; GI, gastrointestinal; ITV, Intervention; MCT, medium-chain triglycerides; NA, not available.

authors observed a superior motor function performance of the intervention group after 7 and 13 weeks, especially for the standing and walking parameters (Table 5).

Omar et al. (2021) [28] studied lactoferrin versus iron hydroxide polymaltose complex for the treatment of iron deficiency anemia. Children between 1 and 10 years old received the interventions for 4 weeks. Biomarkers of anemia were improved in both groups, but the variation in hemoglobin and ferritin was greater in the lactoferrin group. Additionally, children in the lactoferrin group presented a lower incidence of constipation as an adverse effect (Table 5).

The study of Mlinda et al. (2018) [29] investigated whether an educational intervention could improve feeding skills. The intervention consisted of group and individual nutritional education, training of caregivers on positioning during feeding and occupational therapy. Caregivers in the intervention group reported significant improvements in skills, positioning, feeding speed, child involvement during feeding and child-caregiver interaction, in addition to less stress and better child mood during feeding (Table 5).

The results must be interpreted with caution since the overall risk of bias was high for the majority of studies (60.0%). The remaining studies were evaluated with some concerns (40.0%), with no studies presenting low risk of bias. More than 40% of studies presented a high risk of bias or some concerns for three of the five domains evaluated: bias arising from the

**Table 2. Characteristics of the studies evaluating as main outcome plasma concentration of 25-hydroxyvitamin D.**

| Study | Country | N | Age (Years) | Sex | | Functional mobility | Groups (n) | Interventions | Duration of treatment | Outcome studied | Summary of key results |
|---|---|---|---|---|---|---|---|---|---|---|---|
| | | | | F (%) | M (%) | | | | | | |
| Le Roy, 2015 [7] | Chile | 18 | 9.9 (6.2–13.5) | 36.7 | 63.3 | GMFCS level I 3.3% level II 3.3% level V 93.3% | ITV (8) | Single dose of 100,000 IU D3 orally | Single dose | Plasma concentrations of 25OHD | Baseline 25OHD was insufficient in 4/10 of placebo group and 1/8 in the intervention group. After 8 weeks, it remained insufficient in 4/10 in the placebo group and reached the desired levels in all participants of the intervention group. The intervention was not associated with any AE. |
| | | | | | | | Control (10) | Placebo | | | |
| Kilpinen-Loisa, 2007 [25] | Finland | 45 | 9–18 | 43.2 | 56.8 | GMFCS level III 34% level IV 32% level V 34% | ITV (22) | 1000 IU of vitamin D3 orally 5 days/week for 10 weeks | 10 weeks | Plasma concentrations of 25OHD, Calcium homeostasis parameters and bone metabolism | The plasma concentration of 25OHD increased significantly in the supplemented group and decreased in the control group. The intervention was not associated with hypercalcemia or other AE. |
| | | | | | | | Control (23) | Observational (without placebo) | | | |

Note: 25OHD, 25-hydroxyvitamin D; AE, Adverse effects; GMFCS, Gross Motor Function Classification System; ITV, Intervention.

randomization process (13.3% high; 33.3% some concerns), bias due to deviations from intended interventions (6.7% high; 53.3% some concerns), and bias in selection of the reported results (100% some concerns) (Fig 2A).

Ten of the 15 studies (66.7%) did not present a high risk of bias in any domain. Four studies presented some concerns only in one domain (bias in selection of the reported results) [7,12,21,23]; two presented some concerns in two domains [22,28]; and four presented some concerns in three domains [11,20,26,29]. In the five studies with high-risk domains, three presented high risk in only one domain [19,25,27], and two studies presented high risk of bias for two domains [10] (Fig 2B).

## Discussion

The present study synthesized the results and evaluated the risk of bias in studies investigating diet and nutritional interventions on clinical, nutritional and neurodevelopment outcomes in children with CP. Interventions and outcomes varied between the included studies and can be summarized as follows: whey-based or pectin enriched enteral formulas for gastroesophageal reflux; 25-hydroxy-vitamin D supplementation for hypovitaminosis D; supplementation with lipid mixture or diet with high density energy for improvements in anthropometric measures; supplementation with probiotics, prebiotics, symbiotics or magnesium for constipation; nutritional support system for gross-motor function; lactoferrin or iron hydroxide polymaltose for iron deficiency anemia; and educational intervention to improve feeding skills. All studies found positive results and did not report any important adverse outcome. However, evidence is weak since few studies were performed, and the risk of bias is high for the majority of performed studies.

In the studies investigating gastric emptying as an outcome, the proportion of participants with fundoplication was diverse. Included studies with a greater proportion of participants with fundoplication were also those observing more side effects related to faster gastric emptying. Fundoplication is an anti-reflux surgery known for accelerating gastric emptying and may

**Table 3. Characteristics of the studies evaluating as main outcome anthropometric measurements and nutritional status.**

| Study | Country | N | Age (Years) | Sex F (%) | Sex M (%) | Functional mobility | Groups (n) | Interventions | Duration of treatment | Outcome studied | Summary of key results |
|---|---|---|---|---|---|---|---|---|---|---|---|
| Sevilla Paz Soldán, 2018 [11] | Bolivia | 14 | < 5 | 50 | 50 | NA | ITV (7) | 13 vitamins and 6 minerals according to age + 10 mL of lipid mixture containing: Coconut oil (35%), olive oil (35%), marine fish oil (15%), soybean oil (15%). | 6 months | Weight gain and body composition, disability, lipid profile and psychomotor development | The group that received the lipid mixture presented better psychomotor development scores, anthropometric indicators and lipid profile at the end of follow-up |
| | | | | | | | Control (7) | 13 vitamins and 6 minerals according to age | | | |
| Patrick, 1986 [19] | Canada | 15 | 2.8–15.8 | 40 | 60 | NA | ITV (10) | Intensive nasogastric tube-feeding | 5 weeks | Weight gain, triceps skinfold and arm muscle circumference | Intervention increased 10 to 46% of body weight in 4 to 5 weeks with approximately 50% increase in energy intake. The gain in arm muscle circumference and triceps skinfold after the intervention suggests that both lean tissues and fat increased. |
| | | | | | | | Control (5) | Best oral feeding that could be achieved | | | |

Note: ITV, Intervention; NA, not available.

cause dumping syndrome as an adverse effect [30,31]. Therefore, faster gastric emptying in children with fundoplication may be an undesirable outcome, leading to gastrointestinal symptoms, such as gagging, regurgitation, irritability, and pain.

The two studies that evaluated supplementation with vitamin D were performed in cities with low sunlight incidence, Santiago (Chile) and Helsinki (Finland), with latitudes of -33.5 and 60.2, respectively [7,25]. Whereas the closer to zero latitude, the greater the incidence of sunlight, the dosage of vitamin D supplementation to achieve adequate plasma concentrations may vary according to geographical location, being lower for locations near the equator [32].

Regarding the use of magnesium sulfate as an intervention for constipation, the authors raised the limitation that safety data after prolonged treatment for one year were not evaluated [26]. Hypermagnesemia (values above 4 mg/dL) may lead to weakness, nausea, dizziness, confusion, decreased reflexes, worsening confusional state, drowsiness, bladder paralysis, flushing, headache and constipation. Values above 12 mg/dL may cause cardiovascular complications and neurological disorders. Values above 15 mg/dL can induce cardiorespiratory arrest and coma [33]. Therefore, the use of this intervention, especially for long periods, must require close monitoring of magnesium serum levels. Additionally, it may not be recommended for patients with renal disorders, since renal function is crucial for magnesium metabolism [34].

The domains judged as high risk of bias are the most likely cause of spurious associations. The randomization process represented a high risk of bias for two studies. In the study of Kilpinen-Loisa et al. (2007) [25], the allocation sequence was not considered random. The allocation was dependent on the baseline serum concentrations of vitamin D that were arranged in increasing order, and every other child was assigned to receive the supplement. In turn, Leal-Martinez et al. (2020) [27] performed adequate randomization, but baseline results of the motor function scale showed an important difference between groups. Although differences were not statistically significant, the lack of significance may be merely due to the small sample size. They performed an analysis by percentage of evolution to overcome this limitation, but it is still a concern and may be overestimating the effect.

**Table 4. Characteristics of the studies evaluating as main outcome constipation and fecal characteristics.**

| Study | Country | N | Age (Years) | Sex F (%) | Sex M (%) | Functional mobility | Groups (n) | Interventions | Duration of treatment | Outcome studied | Summary of key results |
|---|---|---|---|---|---|---|---|---|---|---|---|
| García-Contreras, 2020 [10] | Mexico | 49 | 1–5 | 38% | 62% | Levels IV and V | Probiotic (12) | 1 × 10e8 cfu of L. reuteri DSM 17938 and 4 g maltodextrin | 28 days | Fecal characteristics (pH, consistency, frequency and microbiota) | Intervention with L. reuteri DSM 17938 and/or inulin agave significantly improved stool characteristics. |
| | | | | | | | Symbiotic (13) | 1 × 10e8 cfu of L. reuteri DSM 17938 and 4 g inulin agave | | | |
| | | | | | | | Prebiotic (13) | 4 g of agave inulin and 5 drops of an oil mixture containing both medium-chain triglycerides and sunflower oil | | | |
| | | | | | | | Placebo (11) | 4 g maltodextrin and 5 drops of oil mixture | | | |
| Hassanein, 2021 [26] | Egypt | 87 | 2–12 | 49% | 51% | GMFCS III to V | ITV (45) | Oral magnesium sulfate solution 4% (4 mg elemental magnesium/mL, 1 mL/kg/day), respecting daily limit (65, 110 and 350 mg/day for 1–3 years, 3–8 years and above 9 years, respectively). | 1 month | Fecal characteristics (consistency, frequency and daily time for evacuation) | The use of oral magnesium sulfate resulted in a significant improvement in constipation scores, consistency and fecal frequency after 1 month, when compared to placebo. |
| | | | | | | | Control (42) | Placebo (Saline Solution) | | | |

Note: GMFCS, Gross Motor Function Classification System; ITV, Intervention.

Garcia-Contreras et al. (2020) [10] had a high risk of bias both due to deviations from the intended intervention and due to missing outcome data. Only 75% of the randomized subjects completed the intervention and were included in the analysis. Moreover, they did not use an appropriate analysis to estimate the effect of assignment to intervention. Patrick et al. (1986) [19] presented a high risk of bias arising from measurement of the outcome. The publication does not present information regarding the blindness of the assessors to the intervention status.

Graham-Parker et al. (2001) [24] is published as an abstract and, therefore, is missing many important information. We tried to contact the authors to obtain more information of the study, without success. It had a high risk of bias for two domains: missing outcome data and measurement of outcome. Abstracts usually do not present information regarding losses of follow-up, consequently we were not able to determine how possible missing outcome data could affect the results. The measurement of outcomes was based on parents' or caregivers' records, which was not considered a reliable method.

Some concerns were raised in all the included studies regarding the bias in selection of the reported results. This occurred because most studies did not present a prespecified analysis plan. To evaluate this domain, the clinical trial register was consulted, and only the study of Garcia-Contreras et al. (2020) [10] did write an analysis plan, but it was not completely followed. Four studies presented some concerns in the overall analysis, instead of low risk, only because of this flaw [7,12,21,23].

None of the included studies performed an intention-to-treat analysis, which is also a major concern to the validity of the results and effectiveness of the interventions in clinical

**Table 5. Characteristics of the studies evaluating others main outcomes as follows: Motor function, laboratory markers of anemia and feeding skills.**

| Study | Country | N | Age (Years) | F (%) | M (%) | Functional mobility | Groups (n) | Interventions | Duration of treatment | Outcome studied | Summary of key results |
|---|---|---|---|---|---|---|---|---|---|---|---|
| Leal-Martínez, 2020 [27] | Mexico | 30 | 4–12 | 40% | 60% | GMFCS III | ITV (10) | Nutritional support system (shake-based diet with functional ingredients, high levels of vegetables, fruits, cereals, roots and fish and supplementation with glutamine, arginine, folic acid, nicotinic acid, zinc, selenium, cholecalciferol, ascorbic acid, spirulina, vegetable protein, PUFAs n-3 and probiotics.) | 13 weeks | Gross-motor function | Children with nutritional support showed superior motor function after 7 and 13 weeks after the beginning of the intervention, especially the parameters "standing up" and "walking". |
| | | | | | | | Control (10) | Diet recommended by WHO) | | | |
| | | | | | | | Follow-up (10) | Monitoring of the usual diet | | | |
| Omar, 2021 [28] | Egypt | 70 | 1–10 | 50% | 50% | GMFCS level I 7.5% level II 18.2% level III 18.2% level IV 19.7% level V 36.4% | ITV (34) | Lactoferrin (oral, 30% bovine iron), 100 mg/day | 4 weeks | Hemoglobin variation, serum iron, biochemical parameters, adherence to therapy, AE | All laboratory markers of anemia were improved with the use of lactoferrin and the polymised ferric hydroxide complex. The variation of hemoglobin and ferritin was significantly higher in children who received lactoferrin, who additionally had a lower incidence of constipation as an AE |
| | | | | | | | Control (32) | Iron hydroxide polymaltose complex (oral, 6 mg/kg/day elemental iron divided into 2 doses) | | | |
| Mlinda, 2018 [29] | Tanzania | 118 | 0–5 | 52.7 | 47.3 | Spastic: 56.4% Quadriplegic: 14.5% Hypotonic: 16.4% Mixed: 12.7% Severity Moderate: 48.2% Severe: 51.8% | ITV (69) | Group and individual nutritional education, caregivers' training on positioning during feeding and occupational therapy for oral motor and functional skills. | 12 months | Feeding skills and caregiver-child interaction during feeding | Caregivers in the intervention group reported significant improvements in positioning skills, feeding speed, child involvement during feeding and child-caregiver interaction, in addition to lower stress and improvement of the child's mood during feeding |
| | | | | | | | Control (49) | Routine general care offered in clinics regularly | | | |

Note: AE, Adverse effects; GMFCS, Gross Motor Function Classification System; ITV, Intervention.

practice. Excluding participants from the analysis according to their adherence to the intervention may leave those who are meant to have a better outcome and damage the unbiased comparison given by randomization [35].

This is the first SLR specifically regarding dietary and nutritional interventions in children with CP. We searched for a broad spectrum of interventions and outcomes in addition to evaluating the risk of bias, which makes this SLR a good referential for health professionals regarding evidence-based practice. Moreover, our search was very sensitive and included gray literature databases, minimizing publication bias. Unfortunately, the results did not allow us to perform a meta-analysis for any intervention/outcome, as the included studies were very heterogeneous.

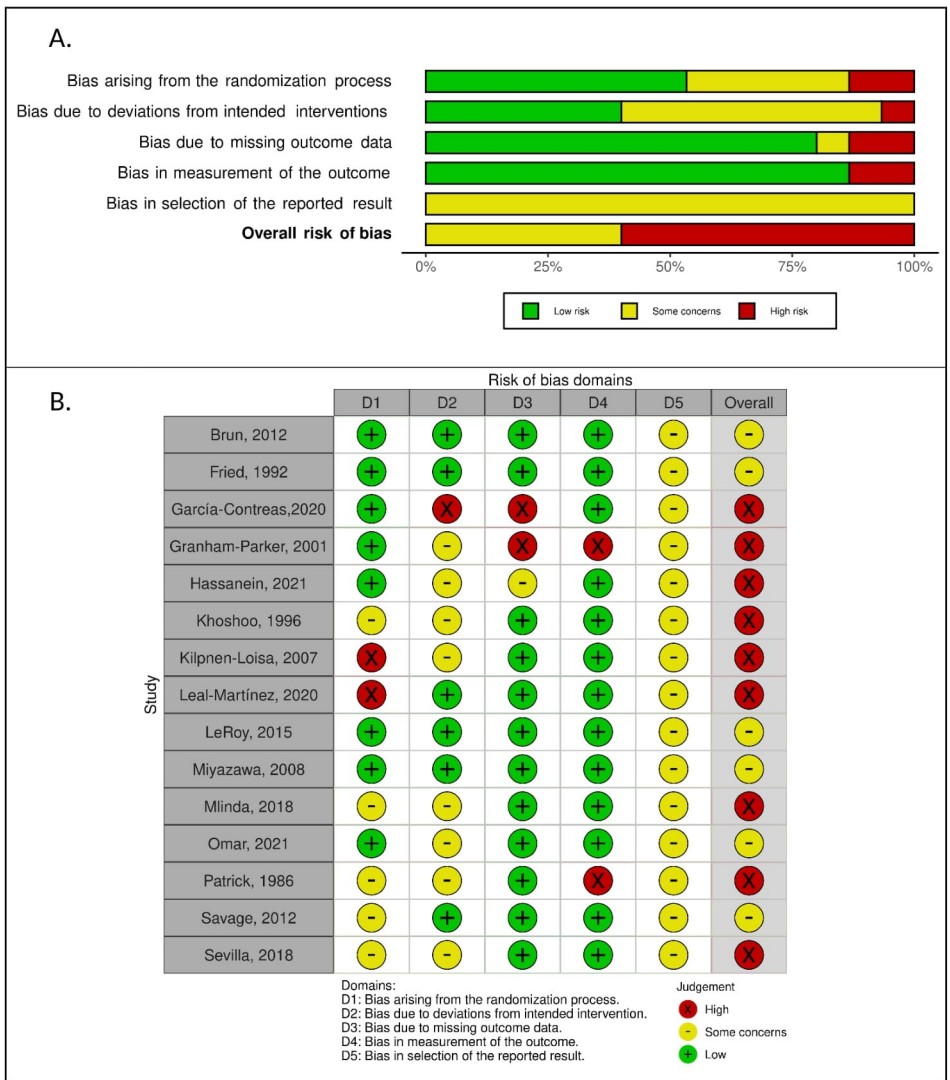

**Fig 2. Risk of bias assessment using Cochrane tool for assessing risk of bias in randomized trials.** A: Risk of bias according to selected domains for each study; B: Proportion of studies with low, high, and unclear risk of bias for each selected domain.

The small number of randomized clinical trials regarding the use of nutritional interventions in children with cerebral palsy together with the promising results found reveals a neglected area with the potential to improve health quality of life for millions of families around the world. Future studies must confirm the results of the interventions evaluated by previous authors using rigorous methods to obtain valid results and, hence, high-quality evidence. Special care must be taken in the randomization process, the intention-to-treat analysis and elaboration and publication of a prespecified analysis plan.

## Supporting information

**S1 Table. Detailed risk of bias assessment using Cochrane tool for assessing risk of bias in randomized trials (RoB 2).**
(XLSX)

**S2 Table. References excluded after full-text evaluation and respective reasons.**
(XLSX)

**S1 Appendix. PRISMA checklist.**
(DOCX)

**S2 Appendix. PROSPERO register details.**
(PDF)

**S3 Appendix. Search summary: Databases, search strategies and number of identified records.**
(DOCX)

**S4 Appendix. Summary of main quantitative results for each study.**
(DOCX)

## Author Contributions

**Conceptualization:** Fernanda Rebelo, Maria Dalva Baker Meio, Saint Clair Gomes Junior.

**Data curation:** Fernanda Rebelo, Isabela Rodrigues Mansur, Teresa Cristina Miglioli.

**Formal analysis:** Fernanda Rebelo, Isabela Rodrigues Mansur, Teresa Cristina Miglioli.

**Funding acquisition:** Fernanda Rebelo, Maria Dalva Baker Meio, Saint Clair Gomes Junior.

**Investigation:** Fernanda Rebelo, Isabela Rodrigues Mansur, Teresa Cristina Miglioli.

**Methodology:** Fernanda Rebelo, Maria Dalva Baker Meio, Saint Clair Gomes Junior.

**Project administration:** Fernanda Rebelo.

**Resources:** Fernanda Rebelo.

**Supervision:** Fernanda Rebelo.

**Writing – original draft:** Fernanda Rebelo.

**Writing – review & editing:** Fernanda Rebelo, Isabela Rodrigues Mansur, Teresa Cristina Miglioli, Maria Dalva Baker Meio, Saint Clair Gomes Junior.

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
