## [Decision Letter · Decision Letter 0]

6 Jun 2022

PONE-D-21-30189Dietary and nutritional interventions in children with cerebral palsy: a systematic literature reviewPLOS ONE

Dear Dr. Rebelo,

Thank you for submitting your manuscript to PLOS ONE. After careful consideration, we feel that it has merit but does not fully meet PLOS ONE’s publication criteria as it currently stands. Therefore, we invite you to submit a revised version of the manuscript that addresses the points raised during the review process.

Please include the following items when submitting your revised manuscript:A rebuttal letter that responds to each point raised by the academic editor and reviewer(s). You should upload this letter as a separate file labeled 'Response to Reviewers'.A marked-up copy of your manuscript that highlights changes made to the original version. You should upload this as a separate file labeled 'Revised Manuscript with Track Changes'.An unmarked version of your revised paper without tracked changes. You should upload this as a separate file labeled 'Manuscript'.

We look forward to receiving your revised manuscript.

Kind regards,

Maria G Grammatikopoulou

Academic Editor

PLOS ONE

Journal Requirements:

Additional Editor Comments (if provided):

Dear authors, two reviewers have worked meticulously on your manuscript and provided their comments. Please reply to the comments and submit your revision,

kind regards

Reviewers' comments:

Reviewer's Responses to Questions

**Comments to the Author**

1. Is the manuscript technically sound, and do the data support the conclusions?

Reviewer #1: Yes

Reviewer #2: Yes

2. Has the statistical analysis been performed appropriately and rigorously? 

Reviewer #1: N/A

Reviewer #2: No

3. Have the authors made all data underlying the findings in their manuscript fully available?

Reviewer #1: Yes

Reviewer #2: Yes

4. Is the manuscript presented in an intelligible fashion and written in standard English?

Reviewer #1: Yes

Reviewer #2: Yes

5. Review Comments to the Author

Reviewer #1: Recommendation: Minor Revisions as suggested on the uploaded PDF file attachment

1. Is the manuscript technically sound, and do the data support the conclusions? Yes the data does support the conclusion to a large extent

2. Has the statistical analysis been performed appropriately and rigorously? N/A. The authors did not perform any statistical analysis because the data collected did not support that.Even-though the intent of the authors was not to conduct a meta-analysis due to the nature of the data presented, a structured narrative data analysis synthesis of included studies could have been conducted using tools such as 'the European Social Research Council guidance on the conduct of narrative synthesis in systematic reviews' or other appropriate tool.

3. Have the authors made all data underlying the findings in their manuscript fully available? Yes, as supplementary material

4. Is the manuscript presented in an intelligible fashion and written in standard English? Yes.

Reviewer #2: Dear Authors

Congratulations on this much expected systematic literature review on such a hot topic.

I really appreciate your effort and your conclusions, which i hope will shape the future research on this specific topic.

I have just a concern: I was not able to find on tables 1-5 the effect estimates and confidence intervals neither the forest plots that were supposed to be submitted as stated in your prism checklist. Could you help me with this?

6. PLOS authors have the option to publish the peer review history of their article (what does this mean?). If published, this will include your full peer review and any attached files.

Reviewer #1: No

Reviewer #2: **Yes: **Thomai Karagiozoglou- Lampoudi

---

## [Author Response · Author response to Decision Letter 0]

20 Jun 2022

Dear Reviewers,

Thank you for your time and effort evaluating our manuscript. We are very grateful for the comments and suggestions, which for sure contributed to enhance the quality of our study.

The answers and considerations for each comment are listed below. All the changes are marked in yellow in the 'Revised Manuscript with Track Changes' file.

Reviewer #1: Recommendation: Minor Revisions as suggested on the uploaded PDF file attachment

The suggestions of on the PDF file are listed below:

I would suggest that authors rather mention the period the search was carried out and leave out the keywords used for the search. Or include both if they haven't exceeded the word limit for the abstract.

The period of the search was included in the abstract. The keywords were excluded due the word limit.

Study (individual RCTs included in the SLR) quality assessment was not clearly and adequately described under the method section.

As described in the ‘Risk-of-bias assessment’ sub-section, the quality of the included studies was evaluated using the RoB 2 tool. To address the reviewers’ concerns, we did a more detailed description of the instrument and data synthesis of risk-of-bias results.

New text (page 7, lines 120-128):

The risk-of-bias assessment was performed using the revised Cochrane tool for assessing risk of bias in randomized trials (RoB 2 tool) [17] in a study level. RoB 2 is a structured tool that evaluate five domains of bias. Each domain, formed by a series of questions, can be judgement as 'Low' or 'High' risk of bias, or can express 'Some concerns'. The assessment was performed by two independent investigators (IRM, TCM). Differences were solved by a third reviewer (FR).

To synthetize the data regarding risk of bias, a summary risk of bias table was generated using the Risk‐of‐bias VISualization (robvis) tool [18]. Results are described as percentual of studies with ‘High risk’ or ‘Some concerns’ for each domain. The domains judged as high risk of bias were discussed to provide reasons for this judgement and recommendations for futures studies. 

Also, data synthesis and statistical analysis was not clearly and adequately described under the method section

We agree and apologize for neglecting this important information. A ‘Data synthesis’ sub-section was incorporated in the manuscript.

New text (page 6, lines 112-118): 

It was not possible to perform a quantitative synthesis due the clinical and methodological heterogeneity of the included studies. For the meta-analysis, at least two studies investigating the same intervention and outcome would be necessary, which did not occur. We performed a structured narrative synthesis based on the Guidance on the Conduct of Narrative Synthesis in Systematic Reviews [16]. The approaches chose for the synthesis, considering the nature of our data, were: textual descriptions of studies; groupings and clusters; and tabulation.

It should read "Search strategy and eligibility criteria". 

We changed the sub-section title as suggested.

If there are clear exclusions (studies that were excluded from the SLR), that should also be mentioned.

The studies excluded were those that did not accomplished the eligibility criteria. As stated in the ‘Results’ section (lines 145 - 146), all the references excluded after full-text evaluation and the respective reasons are listed in S2 Table.

Clearly label the flow chart to reflect the four main domains of the prisma flow chart. Authors labeled only 3 (identification, screening & included). omitted 'Eligibility'

We followed the recommendations from PRISMA 2020, where the flow diagram does not contain the ‘Eligibility’ domain, as you can observe at ‘http://prisma-statement.org/prismastatement/flowdiagram.aspx’.

Define NA at the bottom of the table as you have done for others.

Thank you for the observation. The definition was inserted in the notes of Tables 1 and 3.

Image for figure 2A and 2B are not sharp and clear. Authors should kindly submit a higher resolution image

We will make sure to submit the image with a better resolution.

How the study results was synthesized was not clearly described at the method section. Authors should clearly describe that at the method section.

We agree and apologize for neglecting this important information. A ‘Data synthesis’ sub-section of the ‘Methods’ was incorporated in the manuscript.

New text (page 6, lines 112-118): 

It was not possible to perform a quantitative synthesis due the clinical and methodological heterogeneity of the included studies. For the meta-analysis, at least two studies investigating the same intervention and outcome would be necessary, which did not occur. We performed a structured narrative synthesis based on the Guidance on the Conduct of Narrative Synthesis in Systematic Reviews [16]. The approaches chose for the synthesis, considering the nature of our data, were: textual descriptions of studies; groupings and clusters; and tabulation.

The strength of this study as mentioned from pg 241-246 and the limitations should come or follow after discussing the main findings for each main outcome as suggested by the Prisma guidelines.

The paragraph describing the strengths and limitations was moved to the end of the discussion section (lines 335 - 340).

I was wondering why authors included this study since it was only abstract. Alternatively, authors could have obtained the needed information from the authors of the article.

To avoid publication bias, we did not restrict the included studies on published articles. The referred abstract provided sufficient information to guarantee eligibility. However, it does not contain authors contact information. We searched online for the last author and sent him an e-mail, but unfortunately, we did not have any answer. We added this attempt of contact in the manuscript.

New text (page 16, lines 289 and 290): 

Graham-Parker et al. (2001) [24] is published as an abstract and, therefore, is missing many important information. We tried to contact the authors to obtain more information of the study, without success.

1. Is the manuscript technically sound, and do the data support the conclusions? Yes the data does support the conclusion to a large extent

2. Has the statistical analysis been performed appropriately and rigorously? N/A. The authors did not perform any statistical analysis because the data collected did not support that.Even-though the intent of the authors was not to conduct a meta-analysis due to the nature of the data presented, a structured narrative data analysis synthesis of included studies could have been conducted using tools such as 'the European Social Research Council guidance on the conduct of narrative synthesis in systematic reviews' or other appropriate tool.

As described above, a ‘Data synthesis’ sub-section of the ‘Methods’ was incorporated in the manuscript. We based our synthesis on the ‘Guidance on the Conduct of Narrative Synthesis in Systematic Reviews’ and used the tools that better fitted to our data: textual descriptions, grouping and tabulation.

3. Have the authors made all data underlying the findings in their manuscript fully available? Yes, as supplementary material

4. Is the manuscript presented in an intelligible fashion and written in standard English? Yes.

Reviewer #2: Dear Authors

Congratulations on this much expected systematic literature review on such a hot topic.

I really appreciate your effort and your conclusions, which i hope will shape the future research on this specific topic.

I have just a concern: I was not able to find on tables 1-5 the effect estimates and confidence intervals neither the forest plots that were supposed to be submitted as stated in your prism checklist. Could you help me with this?

Thank you for the positive comments. The effect estimates of the included studies were really missing in our first submission. For the resubmission we added a supplemental material (S4 Appendix) with the statistical tests and main quantitative results for each included study. We also inserted a phrase on the ‘Results’ section to refer to these results.

New text (page 7, lines 153 and 154):

A summary of statistic methods used and main quantitative results for each study can be found in S4 Appendix.

---

## [Decision Letter · Decision Letter 1]

12 Jul 2022

Dietary and nutritional interventions in children with cerebral palsy: a systematic literature review

PONE-D-21-30189R1

Dear Dr. Fernanda Rebelo,

We’re pleased to inform you that your manuscript has been judged scientifically suitable for publication and will be formally accepted for publication once it meets all outstanding technical requirements.

Kind regards,

Maria G Grammatikopoulou

Academic Editor

PLOS ONE

Additional Editor Comments (optional):

Reviewers' comments:

Reviewer's Responses to Questions

**Comments to the Author**

1. If the authors have adequately addressed your comments raised in a previous round of review and you feel that this manuscript is now acceptable for publication, you may indicate that here to bypass the “Comments to the Author” section, enter your conflict of interest statement in the “Confidential to Editor” section, and submit your "Accept" recommendation.

Reviewer #1: All comments have been addressed

Reviewer #2: (No Response)

2. Is the manuscript technically sound, and do the data support the conclusions?

Reviewer #1: Yes

Reviewer #2: (No Response)

3. Has the statistical analysis been performed appropriately and rigorously? 

Reviewer #1: N/A

Reviewer #2: N/A

4. Have the authors made all data underlying the findings in their manuscript fully available?

Reviewer #1: Yes

Reviewer #2: Yes

5. Is the manuscript presented in an intelligible fashion and written in standard English?

Reviewer #1: Yes

Reviewer #2: Yes

6. Review Comments to the Author

Reviewer #1: Authors should make sure that the article is not published anywhere else, all ethics are adhered to

Reviewer #2: Dear authors

thank you for submitting the completed and updated manuscript.

I have to admit that although you followed every step of the relevant methodology I think that due to the multiple questions on the specific topic, and the lack of meta- analysis (which is not your fault) your manuscript is difficult to be accepted as a systematic review.

Nevertheless this manuscript qualifies as an excellent narrative review an I would suggest you to get it resubmitted as such. The revision in this case would be just for you to change the title accordingly, but it represents a major revision

7. PLOS authors have the option to publish the peer review history of their article (what does this mean?). If published, this will include your full peer review and any attached files.

Reviewer #1: **Yes: **Dr Frank Ekow Atta Hayford

Reviewer #2: **Yes: **Thomai Karagiozoglou- Lampoudi

---

## [Editor Report · Acceptance letter]

15 Jul 2022

PONE-D-21-30189R1 

Dietary and nutritional interventions in children with cerebral palsy: a systematic literature review 

Dear Dr. Rebelo:

I'm pleased to inform you that your manuscript has been deemed suitable for publication in PLOS ONE. Congratulations! Your manuscript is now with our production department. 

Kind regards, 

on behalf of

Dr. Maria G Grammatikopoulou 

Academic Editor

PLOS ONE